# The Effectiveness of a Bioactive Healing Abutment as a Local Drug Delivery System to Impact Peri-Implant Mucositis: A Prospective Case Series Study

**DOI:** 10.3390/pharmaceutics15010138

**Published:** 2022-12-30

**Authors:** Piotr Wychowański, Maciej Nowak, Andrzej Miskiewicz, Tadeusz Morawiec, Jarosław Woliński, Zbigniew Kucharski, Pier Carmine Passarelli, Alina Bodnarenko, Michele Antonio Lopez

**Affiliations:** 1Department of Head and Neck and Sensory Organs, Division of Oral Surgery and Implantology, Institute of Clinical Dentistry, Gemelli Foundation for the University Policlinic, Catholic University of the “Sacred Heart”, 00168 Rome, Italy; 2Department of Oral Surgery, Medical University of Gdańsk, 7 Dębinki Street, 80-211 Gdańsk, Poland; 3Department of Periodontology and Oral Diseases, Medical University of Warsaw, 6 Binieckiego Street, 02-097 Warsaw, Poland; 4Department of Oral Surgery, Faculty of Medical Sciences in Zabrze, Medical University of Silesia in Katowice, Pl. Akademicki 17, 41-902 Bytom, Poland; 5Department of Animal Physiology, The Kielanowski Institute of Animal Physiology and Nutrition, Polish Academy of Sciences, 3 Instytucka Street, 05-110 Jabłonna, Poland; 6Department of Prosthodontics, Medical University of Warsaw, 6 St. Binieckiego Street, 02-097 Warsaw, Poland; 7Department of Periodontology and Oral Mucosa Diseases, E. Orzeszkowej 18, 80-208 Gdańsk, Poland; 8Unit of Otolaryngology, University Campus Bio-Medico, Via Álvaro del Portillo, 21, 00128 Rome, Italy

**Keywords:** mucositis, peri-implantitis, clindamycin, bioactive healing abutment, healing abutment, oral microbiota, dysbiosis, dental implants, oral health

## Abstract

Modern dental therapy makes use of prosthetic implant reconstructions, which are supported or retained on dental implants. The most frequent, long-term complications associated with these prosthetic implants include mucositis and peri-implantitis. Since mucositis is the initial inflammation of tissues supporting the dental implant, the management of this condition is thus crucial. The aim of the present study was to assess the effects of the placement of bioactive healing abutment for 48 h, in patients diagnosed with peri-implant mucositis. Moreover, the quantitative and qualitative shift in the bacterial profile of the biofilm present in the peri-implant pockets, was assessed by means of RT-PCR genotyping. Each patient was examined using a commercially available PET test protocol: the first sample was taken upon diagnosis (after which the bioactive healing abutment, with clindamycin at a dose of 30 mg, was used for 48 h and replaced with the prosthetic superstructure used so far by a patient); the second sample was taken two weeks after removal of the bioactive healing abutment. The effects of the intervention were clinically assessed using the PET test after the two weeks. A significant reduction in mucositis was observed following treatment, as measured by periodontal indices: modified Sulcus Bleeding Index—mBI (*p* < 0.001), modified Plaque Index—PLI (r = 0.69, Z= −4.43; *p* < 0.001) and probing depth—PD (Z = −4.61; *p* < 0.001). Significant differences in the occurrence of periopathogenic bacteria were also observed: *Aggregatibacter actinomycetemcomitans* (*p* < 0.014; Z = −2.45; r = 0.38), *Treponema denticola* (*p* < 0.005; Z = −2.83; r = 0.44), *Tannerella forsythia* (*p* < 0.001; Z = −4.47; r = 0.69) and *Porphyromonas gingivalis* (*p* < 0.132; Z = −1.51).

## 1. Introduction

Dental implants are commonly used in prosthodontics to restore single or multiple missing teeth, rehabilitate the stomatognathic system in edentulous patients and to restore the aesthetics of the dental arches [1]. The long-term, single-implant restoration survival rate in a fifteen-year period has been shown to range from between 89–94% [2,3,4].

An increase in awareness, as well as the expectations of patients with regards to oral health quality of life, has forced researchers to develop and use innovative implantological methods. The use of predictable methods including horizontal and vertical bone augmentation, as well as maxillary sinus lift procedures, have allowed for the insertion of implants in patients who were previously ineligible for this type of treatment [5,6]. Thanks to the use of short implants and the advancement of immediate implant techniques, the procedures have become less invasive [7,8]. Atraumatic implantation procedures are not only convincing to previously distrustful patients, but they also enable the use of such techniques in patients with general health burdens [9,10,11].

With the increasing use of dental implants worldwide, the absolute number of failures with regards to implantoprosthetic treatment is also increasing [12]. Depending on whether the failures occur before or after the functional load of the implant, we are dealing with either “early” or “late” failures, respectively. These complications may either be related to the features of the implant itself, the bone or soft tissues adjacent to the implant, or to disorders of the osseointegration process resulting from the patient’s habits and medications [13,14].

Peri-implantitis is the most common cause of implant loss. Long-term observational epidemiological studies estimate that the incidence rates of peri-implant mucositis around dental implants and peri-implantitis range from 40% to 56% and from 20% to 28%, respectively (6th European Workshop on Periodontology, European Academy of Periodontology—Thurgau, Switzerland) [4,15]. Hence, treatment and prevention of peri-implant conditions are the basis of modern treatment in dentistry. The peri-implant microbiota represents a qualitatively inferior but quantitatively superior bacterial ecosystem for some bacterial genera compared to the periodontal microbiota, showing that a progression from healthy state to peri-implantitis causes changes in microbiota composition in the absence of specific disease-causing bacteria. Some studies show an increased concentration of *Prevotella nigrescens* in sites with peri-implantitis, while in periodontitis *Peptostrepstreptococcacee* spp. *Desulfomicrobium orale* were significantly in higher concentrations in periodontitis sites. The microbial community was shown to have evolved from peri-implant health through peri-implant mucositis (with increased concentration of *Porphyromonas gingivalis, Tannerella forsythia and Prevotella intermedia)* to peri-implantitis, characterized by rich microbial communities with an increased concentration of bacteria of genus *Eubacterium* spp. [16] Peri-implant mucositis is defined as inflammation of the soft tissues around the implant, and it is associated with the presence of a bacterial biofilm. Peri-implantitis is an inflammatory condition of the tissues around the implant, characterized by inflammation of the mucosa and progressive bone loss [17]. The most important factor leading to the development of peri-implant diseases is the accumulation of bacterial biofilm. The key factor in this process is the presence of anaerobic gram (-) bacilli such as: *Aggregatibacter actinomycetemcomitans*, *Treponema denticola*, *Tannerella forsythia* and *Porhyromonas gingivalis* [18]. Therefore, disease prevention is aimed at reducing the amount of bacterial biofilm and the number of specific bacteria.

The incidence of peri-implantitis considered at the level of the individual is 18.5% and 12.8% at the level of the implants covered [19]. Risk factors for peri-implantitis include improper oral hygiene, a positive periodontal history (history of periodontitis) and smoking. Diabetes, alcohol consumption and genetic characteristics may also have a negative influence. The peri-implantitis-related microbiota are characterized by a mixed infection by the anaerobic flora. Their composition is comparable to that identified during periodontitis.

At present, there are no generally accepted standards concerning the prevention and treatment of peri-implant diseases. It was proven that preventive general antibiotic therapy reduces the rate of early dental implant failure in healthy patients. General antibiotics, however, do not reduce the risk of infection that can develop many years after implant insertion and prosthetic superstructure delivery [20]. In order to prevent the occurrence of peri-implantitis, various strategies are used, including both the modification of the implants themselves and the use of modern surgical techniques and therapeutic devices [21,22,23,24].

Non-surgical therapy alone does not appear to be effective in a large proportion of cases. Although published cases have shown promising additional benefits when using systemic antibiotics as an adjunct to non-surgical treatments for peri-implantitis, no randomized clinical trials have been conducted to evaluate the effects of systemic antibiotics, including metronidazole, in addition to conservative treatment of peri-implantitis. Moreover, the justification for the use of systemic antibiotics in the periprocedural period during implantation is being questioned due to the possibility of general complications in the patient, the development of drug-resistant strains and the questionable effectiveness in reducing the number of complications [20,25,26]. The current state of medical knowledge prompts the search for modern treatments for peri-implantitis and peri-implant mucositis based on the local application of bacteriostatic and bactericidal drugs [24].

To date, studies aimed at non-surgical methods of eradication or at least remission of peri-implant inflammation have been based on the use of local drug application via implant pockets or medical devices [24]. Implant pocket drug delivery seems to be imprecise in dosage and unpredictable in pharmacodynamics and pharmacokinetics. Medical devices such as bioactive healing abutment can be substantively justified but, so far, their effectiveness has been tested only in vitro [24,27]. Locally delivered antibiotics, together with mechanical debridement, are indicated for non-responding sites of focal infection or in localised recurrent disease. Topical application of the drug, as compared to systemic administration, in the management of periodontal and peri-implant diseases shows a much more favorable effect on the MIC (minimally inhibitory concentration) and is characterized by low-level antibiotic resistance [28]. clindamycin is a bacteriostatic antibiotic of the lincosamide group. It is active against most bacterial strains that play an important role in the etiology of peri-implant and periodontal diseases. When administered in general, it may encounter bacterial resistance and produce some adverse effects. When administered topically, it can reach higher concentrations, which even results in a bactericidal effect, avoiding general health complications. The mechanism of its action is based on the inhibition of bacterial protein biosynthesis, by binding to the 50S ribosome subunit, followed by blocking the A site and 16-L protein [29]. Systemic administration of clindamycin provides a bone concentration of 1/3 that of the serum level. When applied topically, in higher concentrations, clindamycin is bactericidal. Furthermore, systemic use of clindamycin may lead to the development of acute pseudomembranous colitis [30].

The authors searched the PubMed, ProQuest and Web of Science databases using the keywords: clindamycin, bioactive healing abutment, peri-implantitis and peri-implant mucositis and found no publications from human clinical trials. This study is the first attempt to examine the effects of bioactive healing abutment and clindamycin on the reduction of bacteria residing in the peri-implant tissue in humans.

The aim of the study was to conduct a preliminary clinical and microbiological evaluation of the use of an innovative bioactive healing abutment, of its own design, as a tool for the local administration of biologically active agents in the vicinity of a dental implant. The authors wanted to assess the potential impact of clindamycin secreted from this bioactive healing abutment on the condition of periodontal tissues in patients with diagnosed peri-implant mucositis.

## 2. Materials and Methods

### 2.1. General Statements

The study was classified as an observational prospective case series study, according to the taxonomy of Grimes and Schulz clinical trials [31]. To ensure the quality of these studies, the STROBE reporting guidelines recommended by the 7th European Periodontology Workshop were used, and all proposals for a set of outcome domains for clinical research in implant dentistry for peri-implant health (marginal bone level, tissue inflammation—Bleeding On Probing (BOP) with modified Sulcus Bleeding Index (mBI) and probing depth have been investigated [32]. The study was conducted with the consent of the Bioethics Committee at the Medical University of Warsaw no KB/211/2020 and was conducted in accordance with the guidelines of the World Medical Association Declaration of Helsinki—Ethical Principles for Medical Research Involving Human Subjects, Helsinki 2013.

### 2.2. Participants

The participants were 41 generally healthy, non-smoking participants, who were not taking any medications on a regular basis—20 female (48.8%) and 21 male (51.2%)—aged from 39 to 76 years old (M = 55.46; SD = 10.11). Twelve participants received implants in the aesthetic region (29.3%), the other 29 received implants in the premolar and molar areas (70.7%).

### 2.3. Case Qualification

All patients admitted to the study had permanent prosthetic restorations on one to six intraosseous titanium implants in the past. All prosthetic superstructures were made in the form of single crowns screwed on implants or block crowns screwed on implants. All implants were inserted into a fully healed alveolar bone and loaded in a delayed protocol. The time that had passed since the implementation of the permanent prosthetic superstructure ranged from between 26 to 54 months.

A homogenous study group was obtained, consisting of individuals with high socioeconomic status.

The initial recruitment of patients for the study was based on a clinical evaluation performed during the routine follow-up of patients after implant treatment. In most cases (18 persons—60%), patients did not report any subjective symptoms. Peri-implant mucositis was observed in relation to 22 implants (53.66%), the development of which went unnoticed by the patient. All implants were bone level internal connection with a hexagon shape and 3.75 diameter platform.

After the interview was completed, all cases were assessed both periodontally and radiologically. As part of the radiological examination, CBCT (cone beam computed tomography) segmental scans were taken and bone loss was assessed at four points (medial, linguistic, distal and buccal), by two independent examiners. Cases in which bone loss was observed were excluded from the study design.

### 2.4. Periodontal Examination

Periodontal examination included: modified Plaque Index (mPlI), modified Sulcus Bleeding Index (mBI) and Pocket Depth (PD) measurements, as described by Mombelli et. al. [33]. Periodontal examinations were performed by two independent investigators and the results were given as the mean of the studies.

A KerrHawe Click-Probe was used.

Peri-implant mucositis diagnosis was based on the presence of peri-implant inflammation symptoms, including redness, swelling and bleeding on probing without additional bone loss in comparison to the last examination.

The calibration process in periodontology consisted of two parts: examiner alignment and examiner assessment, according to the standard protocol. The mutual agreement was assessed by a pilot study prior to the main periodontal examination. Since the pilot measurements were completed, the obtained discrepancies were at Δ > 1. The κ statistics were used to assess the examiners’ agreement. We calculated that κ >0.8 in all selected to the study indices. Since the two examiners specialized in periodontology, the risk of bias was further lessened.

### 2.5. Study Design

Figure 1 below shows a schematic outline of the study design.

After the diagnosis of peri-implant mucositis, the bacterial profile of the biofilm present in the peri-implant pockets was assessed by means of real-time PCR genotyping (MIP-Pharma). A commercially available PET test kit (MiP Pharma GmbH), in accordance with the described manufacturer protocol we used. Samples were collected by a paper filer and real-time PCR with a fluorescent probe was then performed. In the next step, the prosthetic crown was unscrewed from the implant. The prosthetic superstructure was sent for decontamination and sterilization. The implant socket was then rinsed with saline until any debris was removed and spontaneous bleeding ceased. The bioactive healing abutment (patent no PL238183) described by Iwańczyk, Wychowański et al. [27] was then fixed with the torque of 10 N/cm to the implant affected by the peri-implant mucositis. The BHA was made of grade 5 titanium alloy (Ti6Al4V). The BHA resembled the typical healing abutment and was fixed by screwing to the implant–prosthetic platform. The innovation is bases on the hollow structure inside the abutment that communicate with the peri-implant pockets via release holes. The BHA chamber may be refilled with the active substance thanks to the removable cover mounted on its top.

The chamber of bioactive healing abutment was filled with clindamycin on a collagen carrier and closed with the lid. Figure 2 below shows a schematic outline of the technical data of the bioactive healing abutment.

### 2.6. Preparation of the Insert for the Bioactive Healing Abutment

Dalacin C, 300 mg capsules (Pfizer, Brussels, Belgium), saline and natural collagen—Collacone (Botiss, Zossen, Germany) were used to prepare the cartridge. A total of 30 mg of clindamycin, in the form of Dalacin C 300 mg capsules, was weighed using SBS-LW-200N laboratory balance (Steinberg Systems, Dusseldorf, Germany). The weighed clindamycin was made into a paste using saline on the glass. The prepared paste was used to moisten the fragment of collagen (Collacone, Botiss, Zossen, Germany) located in the bioactive healing abutment chamber. After closure of the lid, the patient was allowed to leave the clinic and then returned after two days.

The bioactive healing abutment was then unscrewed and the implant socket was rinsed with saline and the previously sterilized old prosthetic superstructure was fixed. The second real-time PCR (MIP-Pharma PET) was performed two weeks after prosthetic superstructure restoration. At the same visit, the second periodontal examination was performed including Plaque Index (mPlI), modified Sulcus Bleeding Index (mBI) and Pocket Depth (PD) assessment. All patients who took part in the study remained under constant medical supervision and had the opportunity to obtain appropriate assistance, should the need have arisen.

### 2.7. Statistical Analysis

In order to address the research questions, we carried out statistical analysis on the obtained database using IBM SPSS Statistics v.25 software (Armonk, New York, NY, USA). We calculated basic descriptive statistics using Shapiro–Wilk tests, Wilcoxon’s tests and Mann–Whitney tests. An alpha value threshold of 0.05 (*p* < 0.05) was used. The Shapiro–Wilk test was statistically significant, meaning that the distribution of all variables was significantly different from the normal distribution. Both skewness and kurtosis were high, therefore non-parametric tests were used. Statistical power of the tests used was calculated using the G*Power 3.1.9.4 program. We assumed dependent sample analysis (Wilcoxon tests), moderate effect size, *p* value at the level of 0.95 (α = 0.005) sample size n = 41 and the estimated power of a test was 0.92. As can be observed in the results, in many cases, we observed high effect sizes—if analyzed with such input, the power of a test was 0.999.

## 3. Results

Using the proposed protocol of administration of clindamycin from a bioactive healing abutment in patients with peri-implant mucositis, the authors obtained statistically significant results in relation to the observed some clinical symptoms: level of bleeding gums (mBI); the level of plaque deposit (mPLI), occurrence of both redness and swelling of the gums and probing depths by implants.

Significant differences in the frequency of occurrence of different bacteria strains were observed following treatment. These differences concern the occurrence of: *Aggregatibacter actinomycetemcomitans Treponema denticola*, *Tannerella forsythia* and *Peptostrep. (Micromonas)*
*micros.* The size of all the observed effects was high or very high.

A significantly lower number of bacteria, both total number of bacteria and the number of bacteria in each strain, was observed after the treatment in relation to *Aggregatibacter actinomycetemcomitans*, *Tannerella forsythia*, *Peptostrep. (Micromonas*) *micros*, *Fusobacterium nucleate, Eubacterium nodatum*, *Porphyromonas gingivalis*, *Treponema denticol* and *Capnocytophaga gingivalis* strains.

### 3.1. Bleeding Gums—Modified Sulcus Bleeding Index-mBI

We found a statistically significant frequency reduction in the level of bleeding gums following treatment mBI, Z = −5.39; *p* < 0.001. As can be seen in Table 1, better results were observed in 36 patients while there were no differences in only five of the patients. The size of the observed effect was very high, r = 0.84.

### 3.2. Plaque Deposit—Modified Plaque Index-mPLI

A significant reduction in the level of plaque deposit was observed following treatment mPLI, Z = −4.43; *p* < 0.001. As can be seen in Table 2, mPLI were better in 27 patients, while there were no significant differences observed before and after treatment in 12 of the patients, and slightly worse plaque deposit observed in 2 patients. The size of the observed effect was very high, r = 0.69.

### 3.3. Redness, Swelling and Exudation

A significant reduction in the occurrence of both redness and swelling was observed following in treatment. As can be seen in Table 3, we observed improvement in redness in 41 patients in swelling in 36 patients. The size of both effects were very high. Exudation was not observed upon both measurement I and II.

### 3.4. Depth of Pockets Formed by Implants

A significant reduction in probing depths by implants was observed following treatment, Z = −4.61; *p* < 0.001 (Figure 3). Improvements were observed in 24 of the patients, while there was no change in the remaining 17 patients. The size of the observed effect was very high, r = 0.72.

### 3.5. Occurrence of Bacterial Strains

Significant differences in the frequency of occurrence of different bacteria strains were observed following treatment (Table 4). *Aggregatibacter actinomycetemcomitans* was absent following treatment (at measurement II) in six patients out of the nine patients in which this pathogen was present before treatment (at measurement I). *Treponema denticola*, *Tannerella forsythia* and *Peptostrep. (Micromonas)*
*micros* were all absent following treatment in 8 out of 19 patients, 20 out of 27 patients and *Peptostrep.* in 17 out of 40 patients, respectively. The size of all the observed effects was high or very high. Nearly statistically significant difference in *Eubacterium nodatum* was observed following treatment. This pathogen was absent at measurement II in three out of five patients. Size of this effect was moderate. Other results were not statistically significant. *Porphyromonas gingivalis* was absent following treatment (at in measurement II) in 8 out of 12 patients in which this pathogen was present before treatment (at measurement I). Interestingly, this pathogen was observed in three patients following treatment (at in measurement II), while it was absent in these patients prior to treatment (at measurement I). *Capnocytophaga gingivalis* was absent following treatment in 1 out of 31 patients, while *Prevotella intermedia* was not observed in any of the patients, both before and after treatment.

### 3.6. Number of Bacteria Detected in Peri-Implant Soft Tissues

A significantly lower number of bacteria, both total number of bacteria and the number of bacteria in each strain, was observed after the treatment. The size of the observed effect was very high with regards to the total number of bacteria and for the number of bacteria of *Aggregatibacter actinomycetemcomitans*, *Tannerella forsythia*, *Peptostrep. (Micromonas*) *micros*, *Fusobacterium nucleatum* and *Eubacterium nodatum* strains, high in *Porphyromonas gingivalis*, *Treponema denticola* strains. The effect size was moderate with regards to the number of bacteria of the *Capnocytophaga gingivalis* strain. Results are shown in Table 5.

The total number of bacteria before (measurement I) and after treatment (measurement II) with a bioactive healing abutment containing clindamycin are presented in Figure 4.

### 3.7. Number of Bacteria Detected in Peri-Implant Soft Tissues around Front and Posterior Implants

A significantly lower number of *Fusobacterium nucleatum* and *Capnocytophaga gingivalis* was observed in patients with frontal teeth implants at measurement I (Table 6). The size of the observed effect was moderate for the *Fusobacterium nucleatum* strain and high for the *Capnocytophaga gingivalis* strain.

There were no statistically significant differences in both total number and the number of bacteria per strain. There were also no significant differences in the total number of bacteria or the number of bacteria per strain following treatment (at measurement II).

### 3.8. The Difference in the Number of Bacteria before and after Treatment

We also calculated the difference in the number of bacteria between measurements (i.e., before and after treatment). A significant decrease in the number of *Fusobacterium nucleatum* and *Capnocytophaga gingivalis* was observed in patients with front teeth implants. The size of the observed effects was moderate. Other differences were not statistically significant (Table 7).

### 3.9. The Case Presentation

The case study referring to the peri-implant mucositis of 12 implant treatments in the described research scheme is shown in Figure 5 and Figure 6.

## 4. Discussion

This is the first study to present a novel method of mucositis treatment in patients after implantoprosthetic proceedings. To date, there has been no constructed study on animals concerning the use of the bioactive healing abutment containing an antibiotic. The differences in bacterial contamination between animals and humans seems to constitute an insurmountable obstacle at present. An animal’s tooth size is significantly different to that of a human, and the diet and chewing function seems to be the next obstacle to providing a reliable animal model to study the action of the bioactive healing abutment. In order to obtain the most reliable data, the study was conducted on humans after obtaining approval from the bioethics committee, as a follow-up of the in vitro study [26]. 

The use of the bioactive healing abutment resulted in a reduction in the levels of clinical indices of inflammation, measured using mBI, mPI and PD indicators. Moreover, a decrease in redness and swelling of the gums around the implants was observed. In the genetic study, a significant reduction in the total number of bacteria was observed, as well as a significant reduction in the number of *Aggregatibacter actinomycetemcomitans* (Aa), *Treponema denticola* (Td), *Tanerella forsythia* (Tf) and *Peptostreptococcus micros* (Pm). A decrease in the number of *Porphyromonas gingivalis* (Pg) was also observed, although it was not significant. Most of these bacteria belong to the red complex, according to Socransky [34]. The presence of these bacteria is associated with inflammation of periodontal tissues, which clinically manifests itself as bleeding on probing. The red complex bacteria produce specific invasins and are able to relocate, which enables them to penetrate the gum tissue and makes them difficult to remove through standard debridement methods. In addition, gram (-) bacteria, with Aa being one of them, activate the inflammasome complex, which through the action of NLRP proteins leads to an increase in the secretion of pro-inflammatory cytokines [35]. Hence, there is a theoretical basis for the use of clindamycin in bioactive healing abutment in order to reduce the number of bacteria and in doing so, the local inflammatory reaction [36]. Practical studies, with clindamycin’s use as a pharmacologically active agent placed in bioactive healing abutment have been carried out only in vitro [27].

The causes of mucositis and peri-implantitis are diverse. The most common causes include improper hygiene measures, residual bacterial biofilm around the implants, reduced width of keratinized gums, occlusal overload and local factors related to the design of implants, including material imperfections such as microleakage and misfit of the abutment [37]. In the long run, most of the above-mentioned factors lead to the colonization of the implant structures by particularly pathogenic bacteria such as Aa and Pg. The bacterial biofilm leads to the development of mucositis. If left untreated, the inflammation spreads to the bone that is supporting the implant and, consequently, peri-implantitis develops [38]. Many previous efforts to prevent peri-implantitis were based on the modification of the implant surface by: modified sandblasting and an acid etching technique (mod-SLA), coating with hydroxyapatite or polyglycolactonic acid (PLGA) and covering the implant surface with recombinant human bone morphogenetic protein (rhBMP-2) solution. These methods influenced the process of osseointegration of the implant, which is crucial in the healing phase, but did not constitute a line of defense against bacterial colonization, and thus did not protect against the development of mucositis or peri-implantitis.

Earlier attempts related to the local use of antibiotics in the treatment of peri-implantitis involved performing an open-flap debridement, and then a drug was applied to the prepared surface of the implant and to the position of the bone lesion. The results obtained indicated a reduction in inflammation, which led to a reduction in the loss of dental implants due to peri-implantitis [39]. Peri-implantitis is an advanced form of inflammation of the tissues around the implant. In modern treatment, we rely on the early diagnosis of mucositis and the implementation of mucositis treatment methods, as it is a reversible inflammation of soft tissues that does not yet affect bone structure. Therefore, controlling mucositis is a form of peri-implantitis prevention. The treatment of peri-implantitis is long, costly, burdensome for the patient and the prognosis is uncertain. Moreover, the effectiveness of regenerative methods used in the treatment of peri-implantitis depends on many factors, such as: age, disease burden, general medications used, and on local factors including bone defect conformation and tobacco smoking. Hence, regimens for the treatment of mucositis and the prevention of peri-implantitis have been developed for years.

In the year 2000, the cumulative interceptive supportive treatment (CIST) protocol was published [40]. In the case of mucositis, this protocol recommended mechanical debridement and antiseptic cleansing. Only in the case of pocket depths (PD) of >5.0 mm and radiologically confirmed bone loss of 2.0 mm, local or general antibiotic therapy and/or regenerative surgery were advised. The described CIST protocol did not assume the use of an antibiotic in the treatment until the condition developed into what we now call peri-implantitis. Moreover, in the case of inflammation of the soft tissues around the implants (PD ≤ 3.0 mm and BOP (+) positive), only local measures of the peri-implant pocket treatment were implemented. To our knowledge, bacteria such as Aa, Pg, Tf and Td secrete a number of collagenases and invasins, the activity of which leads to the lysis of extracellular matrix, and finally the penetration of these bacteria into the gum tissue. Clindamycin is characterized by a high rate of gingival infiltration [41]. It is active against gram (+) and (-) bacteria, especially against anaerobic bacteria present in the tissue of infected gums. Topical application of clindamycin promotes rapid achievement of high active concentrations at the site of administration and reduces the risk of developing antibiotic resistance. Moreover, there is no risk of developing *C. difficile* colitis, which is common when clindamycin is administered orally.

In most cases, the authors noted a significant decrease in the number of individual bacterial strains in the peri-implant pockets of patients after the application of clindamycin with BHA. The registered growth of *P. gingivalis* in three tested people may be the result of some hygienic negligence, pharmacodynamics of clindamycin release or drug resistance of bacterial strains. The resistance of some strains of periopatogens on clindamycin should be considered [42].

The significant advantages of using the described method of mucositis treatment include the fact that the described method of treatment is not bothersome for the patient. The application of the drug in bioactive healing abutment takes place during one dental appointment. In addition, there is no need for surgery, and fixing the bioactive healing abutment itself does not require anesthesia or incision of the gum. The procedure is quick and relatively cheap. The most important feature obtained in the proposed treatment protocol is the long-lasting elimination of periopathogenic bacteria, as the second measurement was performed two weeks after the removal of the bioactive healing abutment. Limitations of the study include the inability to control drug release over time, and the lack of established guidelines for oral hygiene in patients with mucositis.

Modern strategies for the clinical and microbiological management of the oral microbiota include personalized protocols with a proactive approach. These algorithms include a combination of minimally invasive instrumentation, to avoid tissue reorganization, and the topical use of probiotics and paraprobiotics to enhance these treatments. Proactive actions act only on the risk factors, and rebalancing the bacterial biota present within the gingival sulcus and cervical fluid. These strategies are willingly accepted by the patients due to their atraumatic character as well as avoidance of the use of antibiotic. The proposed bioactive healing abutment may be a useful active agent delivery toll to peri-implant sulcus in these modern periodontal treatment strategies [43,44,45,46].

The release time of the active agent varies with the drug used, the carrier used as well as the number and size of releasing holes and the clinical conditions. In the presented in-vivo study, periodontium is an important factor influencing the release of antibiotic. The probing depth of the peri-implant pocket, periodontal phenotype and gingival fluid flow, as an inflammatory exudate may be important variables. The above factors constitute the further limitations of the study and the need for in vitro and in vivo study to optimize the drug release curve.

Another limitation of our study was the performance of only bacteriological and clinical examinations of the peri-implant tissues, without the examination of cytokine and enzyme levels. Our results, therefore, reflect the sum of the actions (bacteriostatic, bactericidal and anti-inflammatory) of clindamycin on the peri-implant tissues and do not assess the contribution of individual components to the final effect.

The case series is the first report to present the method. It has to be proven by carrying out a research design that includes the control group as a cohort study or even experimental study. Hence, there is a need for further research on a larger group of patients, longer follow-ups and other types of research projects.

## 5. Conclusions

The use of a bioactive healing abutment decreases of the numbers of anaerobic gram (-) bacilli, which play a key role in the development of peri-implant mucositis and in the etiology of peri-implantitis, thus reducing the clinical parameters of peri-implant mucositis in a short time period.

Systematic monitoring of patients with dental implants can help detect the asymptomatic onset of peri-implant inflammation and BHA may in the future be a valuable tool in the hands of medical practitioners to treat this pathology.

Further research should be conducted in order to confirm the results obtained in the present study, as well as the faith of implants especially due to the recurrence of the peri-implant mucositis and the risk of peri-implantitis development around these implants.

The bioactive healing abutment may be a useful drug/active substance delivery toll to perform minimally invasive oral microbiome management protocols in modern periodontal proactive action strategies to balance oral dysbiosis.

## Figures and Tables

**Figure 1 pharmaceutics-15-00138-f001:**
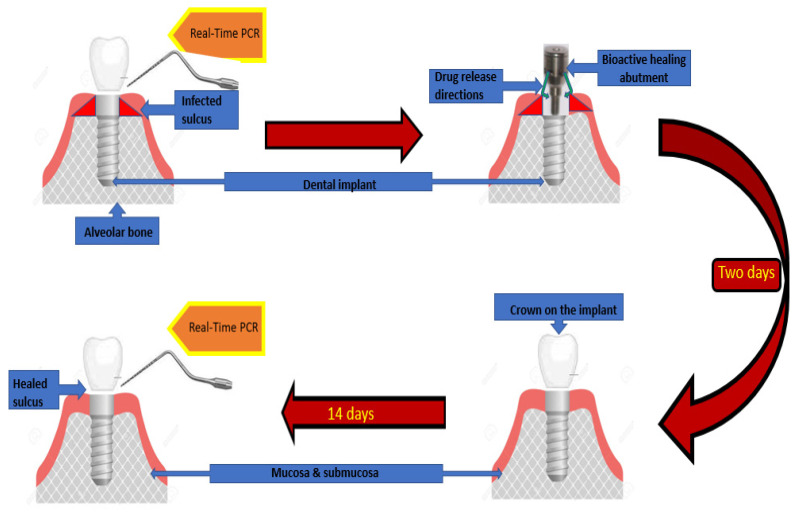
The research study scheme design.

**Figure 2 pharmaceutics-15-00138-f002:**
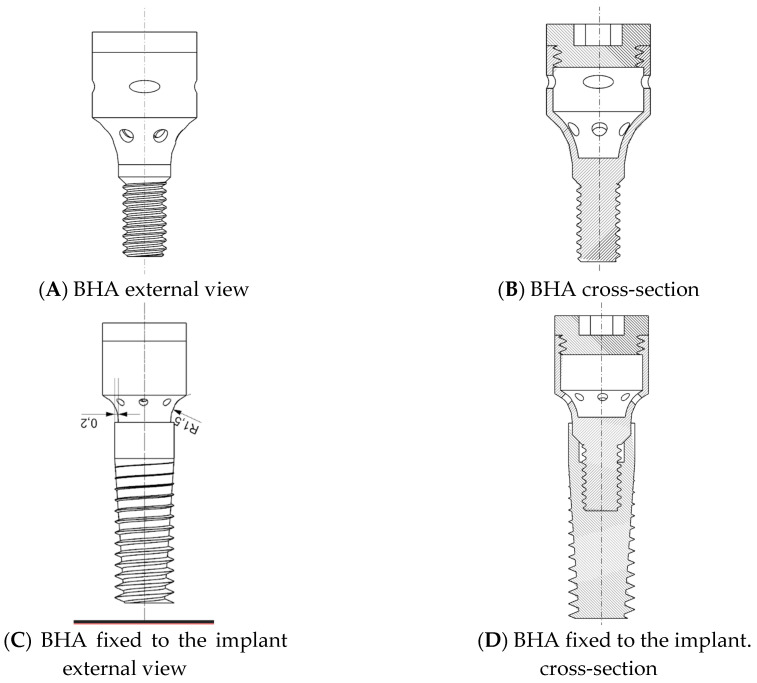
Technical data of the bioactive healing abutment scheme.

**Figure 3 pharmaceutics-15-00138-f003:**
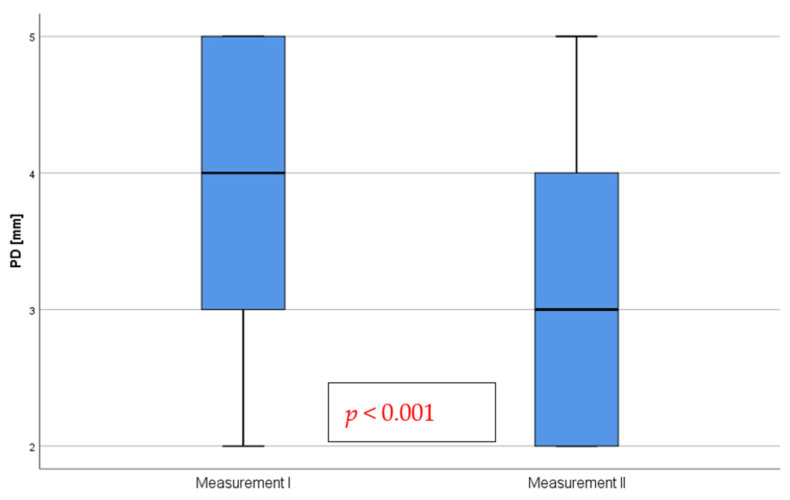
Depth of the pockets formed by the dental implants before (measurement I) and after treatment (measurement II) with a bioactive healing abutment containing clindamycin.

**Figure 4 pharmaceutics-15-00138-f004:**
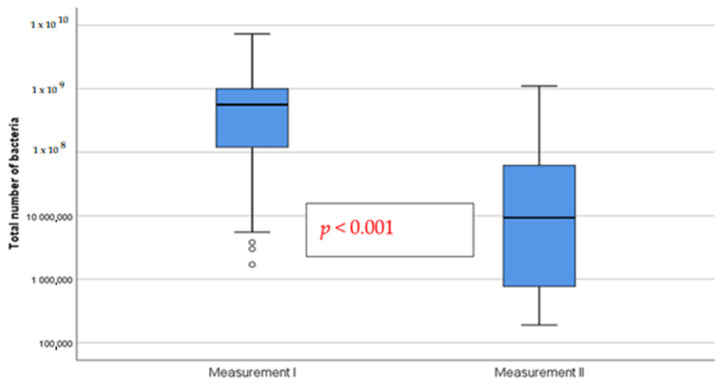
Total number of implant gingival pockets of patients before (measurement I) and after treatment (measurement II) with a bioactive healing abutment containing clindamycin.

**Figure 5 pharmaceutics-15-00138-f005:**
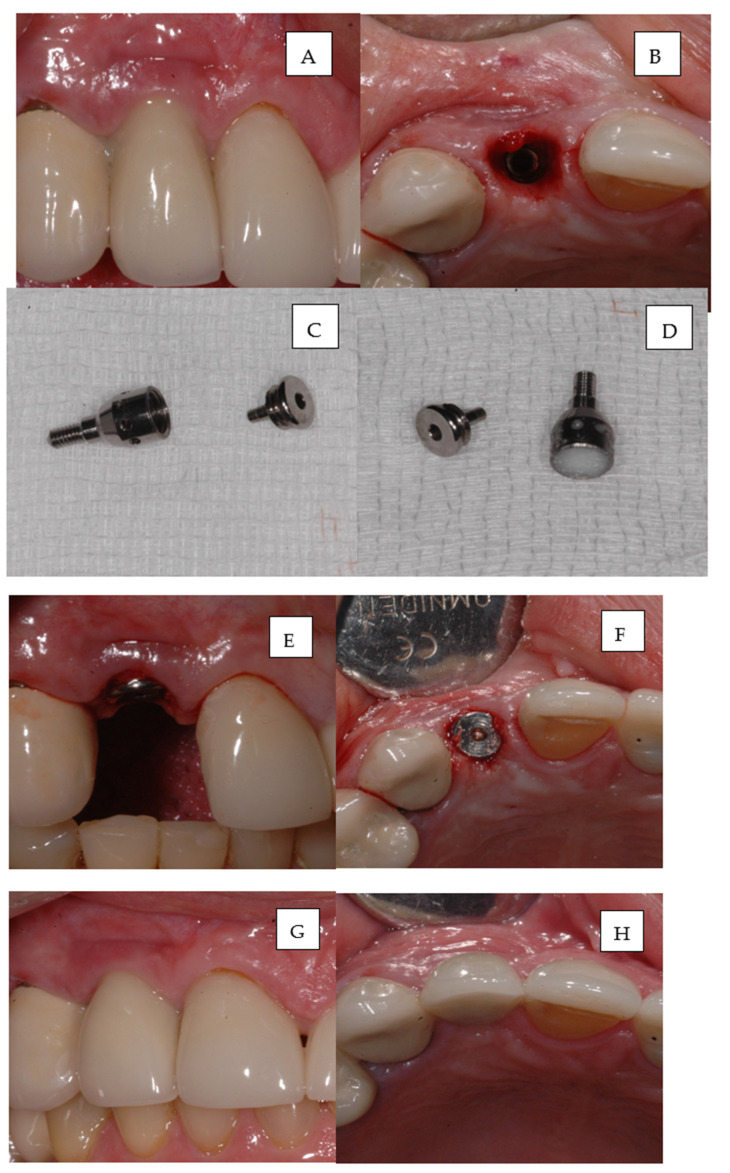
Clinical case. bioactive healing abutment (BHA) use for the treatment of 12 peri-implant mucositis. (**A**). Before treatment; (**B**). Crown unscrewed; (**C**). BHA; (**D**). BHA chamber filled with clindamycin. (**E**). BHA fixed on implant, treatment starting point; (**F**) BHA fixed on implant after 2 days. (**G**,**H**) 2 weeks after treatment with BHA with clindamycin.

**Figure 6 pharmaceutics-15-00138-f006:**
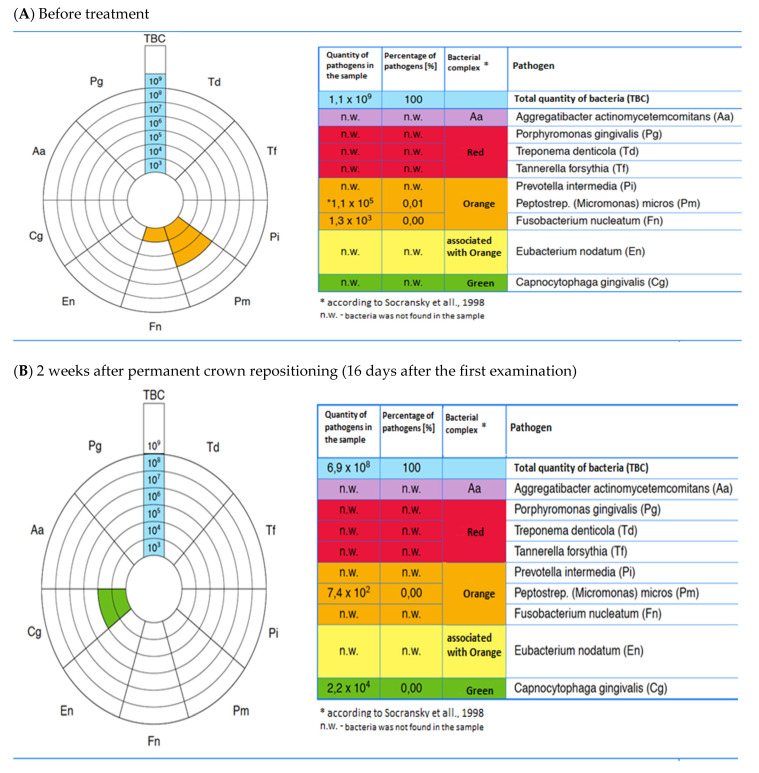
Bacterial profile of the biofilm present in the peri-implant pocket in th clinical case referred to in Figure 2, before (**A**) and after (**B**) treatment with a bioactive healing abutment (BHA) containing clindamycin. Real-time PCR study.

**Table 1 pharmaceutics-15-00138-t001:** The incidence of bleeding gums before (measurement I) and after treatment (measurement II) with a bioactive healing abutment containing clindamycin.

			Measurement I mBI			
			Absent	Blood Point	Bloody Line	Profuse Bleeding
Measurement II mBI	absent	*n*	0	7	6	1
		%	0.00%	17.10%	14.60%	2.40%
	blood point	*n*	0	5	10	10
		%	0.00%	12.20%	24.40%	24.0%
	bloody line	*n*	0	0	0	2
		%	0.00%	0.00%	0.00%	4.90%

**Table 2 pharmaceutics-15-00138-t002:** Plaque deposit—mPLI before (measurement I) and after treatment (measurement II) with a bioactive healing abutment containing clindamycin.

				Measurement I mPLI		
			Absent	Thin Layer of Plaque	Moderate Layer of Plaque	Abundant Plaque Deposits
Measurement II	absent	*n*	3	5	6	4
		%	7.30%	12.20%	14.60%	9.80%
	thin layer of plaque	*n*	2	9	12	0
		%	4.90%	22.00%	29.30%	0.00%

**Table 3 pharmaceutics-15-00138-t003:** Redness and swelling before (measurement I) and after treatment (measurement II) with a bioactive healing abutment containing clindamycin.

				Measurement I		
				Redness		
				Absent	Present	
Measurement II	Redness	Absent	*n*	0	41	*Z* = −6.40 *p* < 0.001 *r* = 1
			%	0%	100%	
		Present	*n*	0	0	
			%	0%	0%	
				Swelling		
				Absent	Present	
	Swelling	Absent	*n*	1	36	*Z* = −6.00 *p* < 0.001 *r* = 0.94
			%	2.4%	87.8%	
		Present	*n*	0	4	
			%	0%	9.8%	

**Table 4 pharmaceutics-15-00138-t004:** Occurrence of bacteria strains before (measurement I) and after treatment (measurement II) with a bioactive healing abutment containing clindamycin.

				Measurement I		
				Aggregatibacter actinomycetemcomitans		
				Absent	Present	
Measurement II	Aa	Absent	*n*	32	6	** *Z* ** **= −2.45** ***p* = 0.014** ***r* = 0.38**
			%	78.00%	14.60%	
		Present	*n*	0	3	
			%	0.00%	7.30%	
				Porphyromonas gingivalis		
				Absent	Present	
	Pg	Absent	*n*	25	8	*Z* = −1.51 *p* = 0.132
			%	61.00%	19.50%	
		Present	*n*	3	5	
			%	7.30%	12.20%	
				Treponema denticola		
				Absent	Present	
	Td	Absent	*n*	22	8	** *Z* ** **= −2.83** ***p* = 0.005** ***r* = 0.44**
			%	53.70%	19.50%	
		Present	*n*	0	11	
			%	0.00%	26.80%	
				Tannerella forsythia		
				Absent	Present	
	Tf	Absent	*n*	14	20	** *Z* ** **= −4.47** ***p* < 0.001** ***r* = 0.69**
			%	34.10%	48.80%	
		Present	*n*	0	7	
			%	0.00%	17.10%	
				Peptostrep. (Micromonas) micros		
				Absent	Present	
	Pm	Absent	*n*	1	17	** *Z* ** **= −4.12** ***p* < 0.001** ***r* = 0.64**
			%	2.40%	41.50%	
		Present	*n*	0	23	
			%	0.00%	56.10%	
				Fusobacterium nucleatum		
				Absent	Present	
	Fn	Absent	*n*	4	15	** *Z* ** **= −3.87** ***p* < 0.001** ***r* = 0.60**
			%	9.80%	36.60%	
		Present	*n*	0	22	
			%	0.00%	53.70%	
				Eubacterium nodatum		
				Absent	Present	
	En	Absent	*n*	36	3	*Z* = −1.73 *p* = 0.083 *r* = 0.27
			%	87.80%	7.30%	
		Present	*n*	0	2	
			%	0.00%	4.90%	
				Capnocytophaga gingivalis		
				Absent	Present	
	Cg	Absent	*n*	7	1	*Z* = −1.00 *p* = 0.317
			%	17.10%	2.40%	
		Present	*n*	3	30	
			%	7.30%	73.20%	

**Table 5 pharmaceutics-15-00138-t005:** Number of bacteria per strain in implants gingival pockets of patients before (measurement I) and after treatment (measurement II) with a bioactive healing abutment containing clindamycin.

	Measurement I			Measurement II					
	*Me*	*IQR*	*Min-Max*	*Me*	*IQR*	*Min-Max*	*Z*	*p*	*r*
**Total number of bacteria**	560,000,000	962,000,000	7,298,300,000	9,300,000	62,825,000	1,099,810,000	−4.61	**<0.001**	0.72
**Aggregatibacter actinomycetemcomitans**	0	0	5300	0	0	210	−5.48	**<0.001**	0.86
**Porphyromonas gingivalis**	0	530	3,300,000	0	0	430	−2.67	**0.008**	0.42
**Treponema denticola**	0	7850	132,000	0	25	23,000	−3.03	**0.002**	0.47
**Tannerella forsythia**	450	3150	390,000	0	0	24,000	−3.82	**<0.001**	0.60
**Peptostrep. (Micromonas) micros**	19,000	183,625	930,000	140	390	24,000	−4.54	**<0.001**	0.71
**Fusobacterium nucleatum**	20,000	47,700	430,000	110	210	26,000	−5.42	**<0.001**	0.85
**Eubacterium nodatum**	0	0	7700	0	0	130	−5.30	**<0.001**	0.83
**Capnocytophaga gingivalis**	6800	169,810	590,000	720	1705	29,000	−2.02	**0.043**	0.32

**Table 6 pharmaceutics-15-00138-t006:** Number of bacteria detected in peri-implant soft tissues of patients, depending on the position of the implant, before (measurement I) and after treatment (measurement II) with a bioactive healing abutment containing clindamycin.

	Front Teeth (*n* = 12)			Side Teeth (*n* = 29)					
	*Me*	*IQR*	*Min-Max*	*Me*	*IQR*	*Min-Max*	*Z*	*p*	*r*
**Measurement I**									
**Total numer of bacteria**	545,000,000	767,175,000	7,298,300,000	560,000,000	1,162 × 10^9^	6,397,000,000	−0.23	0.819	0.04
**Aggregatibacter actinomycetemcomitans**	0	0	2300	0	45	5300	−0.63	0.527	0.10
**Porphyromonas gingivalis**	0	2607.5	3,300,000	0	460	2,700,000	−0.26	0.795	0.04
**Treponema denticola**	1,500	14,500	95,000	0	5550	132,000	−0.45	0.651	0.07
**Tannerella forsythia**	325	6375	390,000	590	11,900	330,000	−0.54	0.588	0.08
**Peptostrep. (Micromonas) micros**	23,000	275,200	839,650	19,000	129,360	930,000	−0.98	0.330	0.15
**Fusobacterium nucleatum**	46,500	36,975	430,000	6500	46,150	77,000	−2.07	**0.039**	0.32
**Eubacterium nodatum**	0	0	7700	0	0	5700	−0.61	0.545	0.09
**Capnocytophaga gingivalis**	27,000	465,100	509,180	1000	76,000	590,000	−2.50	**0.013**	0.39
**Measurement II**									
**Total numer of bacteria**	18,000,000	116,547,500	1,099,810,000	6,300,000	50,465,000	689,770,000	−0.92	0.359	0.14
**Aggregatibacter actinomycetemcomitans**	0	0	0	0	0	210	−1.14	0.253	0.18
**Porphyromonas gingivalis**	0	362.5	430	0	0	310	−1.74	0.082	0.27
**Treponema denticola**	0	0	30	0	45	23,000	−1.18	0.240	0.18
**Tannerella forsythia**	0	0	290	0	0	24,000	−0.04	0.965	0.01
**Peptostrep. (Micromonas) micros**	0	385	2100	200	430	24,000	−0.94	0.346	0.15
**Fusobacterium nucleatum**	115	207.5	260	90	210	26,000	−0.02	0.988	0.00
**Eubacterium nodatum**	0	0	130	0	0	100	−0.69	0.490	0.11
**Capnocytophaga gingivalis**	670	2080	4300	720	1785	29,000	−0.07	0.943	0.01

**Table 7 pharmaceutics-15-00138-t007:** The difference in the number of bacteria between measurements (i.e., before and after treatment with a bioactive healing abutment containing clindamycin) in patients with front and side teeth implants.

	Front Teeth (*n* = 12)			Side Teeth (*n* = 29)					
	*Me*	*IQR*	*Min-Max*	*Me*	*IQR*	*Min-Max*	*Z*	*p*	*r*
Total numer of bacteria	474,615,000	729,090,000	6,210,900,000	420,700,000	1057 × 10^9^	6,329,410,000	−0.34	0.731	0.05
Aggregatibacter actinomycetemcomitans	0	0	2300	0	25	5,090	−0.63	0.527	0.10
Porphyromonas gingivalis	0	2607.5	3,300,090	0	460	2,699,880	−0.31	0.757	0.05
Treponema denticola	1485	14,500	95,000	0	3145	132,000	−0.69	0.493	0.11
Tannerella forsythia	325	637.5	389,710	590	4400	329,790	−0.54	0.588	0.08
Peptostrep. (Micromonas) micros	23,000	274,382.5	839,460	17,000	129,180	929,990	−1.22	0.223	0.19
Fusobacterium nucleatum	46,305	3697.5	429,790	4610	45,995	76,790	−2.14	**0.033**	0.33
Eubacterium nodatum	0	0	7570	0	0	5700	−0.60	0.545	0.09
Capnocytophaga gingivalis	23,500	465,045	510,160	250	71,190	618,620	−2.24	**0.025**	0.35

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
