# Peer review of "The Effectiveness of a Bioactive Healing Abutment as a Local Drug Delivery System to Impact Peri-Implant Mucositis: A Prospective Case Series Study"

_pharmaceutics, 2022, doi:10.3390/pharmaceutics15010138_

Round 1
Reviewer 1 Report
The manuscript shows promising antibacterial effects of a Bioactive Healing Abutment against periodontal pathogens. Therefore, I recommend this article for publication after considering the following modifications.
- Figure 1, please increase the font sizes in blue boxes to increase the understanding.
- Swelling and redness decreased in patients after having BHA coated with Clindamycin. Was this effect because of the anti-inflammatory properties of Clindamycin, or was there a reduced bacterial number that led to a decrease in these inflammatory signs?
- Line 352-354, the authors report an increase in P. gingivalis in three patients despite having BHA. What was reason behind an increase in bacterial number? Was there any role of food behind this? Hopefully, there was a controlled release of Clindamycin from BHA in this situation.
- Line 353, remove this before the pathogen.
- Authors are advised to provide a general recommendation about the use of BHA. Considering their benefits and antibacterial activities, should BHA be routinely applied in patients?
- BHA did not show strong effect against some bacteria, do authors suggest to include a combination of antibiotics in BHA instead of using a single antibiotic?
- Please make references, 5,11,13,31, and 32 similar to other references.
Reviewer 2 Report
Manuscript of considerable interest for the dental sector, a major revision is required before evaluating a possible publication
Abstract, highlight statistically significant data
Few keywords: add specific ones
Introduction: how does the oral microbiota change in the implant patient? Add reference from Prof Scribante
10.3390/app12073250Materials and methods: how was the sample size calculated?
Results: very confusing, highlight statistically significant data
Discussion: Add proactive action using natural substances to reduce the incidence of pathogenic microorganisms as future goals
10.3390/microorganisms10040675
Conclusion; add proactive action
Round 2
Reviewer 2 Report
the manuscript was properly revised based on the comments that were suggested in the first session